Cryopreservation of lumpfish Cyclopterus lumpus (Linnaeus, 1758) milt

Norðberg Gunnvør gunnvorj@fiskaaling.fo
Johannesen Asa
Arge Regin
Fiskaaling, Aquacultural Research Station of the Faroes , við Áir, Hvalvík , Faroe Islands
Hamre Kristin
Electronic publication date: 2015 Jun 4
Publication date: 2015
Volume: 3
Electronic Location ID: e1003
Received 2014 Dec 22; Accepted 2015 May 16
Copyright: © 2015 Norðberg et al.
Copyright year: 2015
Copyright holder: Norðberg et al.
License: This is an open access article distributed under the terms of the Creative Commons Attribution License, which permits unrestricted use, distribution, reproduction and adaptation in any medium and for any purpose provided that it is properly attributed. For attribution, the original author(s), title, publication source (PeerJ) and either DOI or URL of the article must be cited.
License URL: https://creativecommons.org/licenses/by/4.0/

Keywords: Lumpfish, Cyclopterus lumpus, Cryopreservation, Milt motility, CASA

Funding: This work was funded by the Faroese fish farmer’s association as part of a collaborative project aimed at improving lumpfish breeding practices. The funders had no role in study design, data collection and analysis, decision to publish, or preparation of the manuscript.

==============================
This study has established a successful protocol to cryopreserve lumpfish Cyclopterus lumpus (Linnaeus, 1758) milt. Three cryosolutions were tested based on Mounib’s medium; the original medium including reduced l-glutathione (GSH), the basic sucrose and potassium bicarbonate medium without GSH, or with hen’s egg yolk (EY). Dimethyl sulphoxide (DMSO) was used as the cryoprotectant along with all three diluents in a 1–2 dilution. Cryopreservation was performed with the mentioned cryosolutions at two freezing rates. Motility percentages of spermatozoa were evaluated using ImageJ with a computer assisted sperm analyzer (CASA) plug-in. Findings revealed that spermatozoa cryopreserved in Mounib’s medium without GSH had a post-thaw motility score of 6.4 percentage points (pp) higher than those in the original Mounib’s medium, and an addition of EY to the modified Mounib’s medium lowered the post-thaw motility score by 19.3 pp. The difference in motility between both freezing rates was 13.0 pp, and samples cryopreserved on a 4.8 cm high tray resulted in a better post-thaw motility score. On average, cryopreserved milt had a 24.1 pp lower post-thaw motility score than fresh milt. There was no significant difference in fertilisation success between cryopreserved and fresh milt. Cryopreservation of lumpfish milt has, to our knowledge, never been successfully carried out before. The established protocol will be a main contributing factor in a stable production of lumpfish juveniles in future.

Introduction

Cryopreservation is an effective method for long-term storage of viable spermatozoa in fish (Blaxter, 1953). This technique offers several benefits including artificial fertilisation, which allows for efficient use of milt. This is especially useful in species from which milt samples are difficult to obtain (Ohta & Izawa, 1996), or when limited volumes can be stripped (Clearwater & Crim, 1995). Other advantages of the technique include the option to preserve declining stocks (Tian et al., 2008) and to retain genetic variability in broodstocks (Suquet et al., 2000).

Milt cryopreservation has been well established for some freshwater fish species belonging to families of salmonids (Billard, 1992), cyprinids (Billard et al., 1995) and siluroids (Legendre, Linhart & Billard, 1996). Cryopreservation efforts for marine fish species were ongoing already in the 80’s (Bolla, Holmefjord & Refstie, 1987), and (several) successful cryopreservation protocols have been defined (Suquet et al., 2000), for e.g., Atlantic haddock (Melanogrammus aeglefinus) (Rideout, Trippel & Litvak, 2004), Olive flounder (Paralichthys olivaceus) (Zhang et al., 2003), Brazillian flounder (Paralichthys brasiliensis) (Lanes et al., 2008), and Atlantic halibut (Hippoglossus hippoglossus) (Ding et al., 2011). However, it is a challenging task to determine an optimal protocol for cryopreservation of milt for any particular fish species as diluent and cryoprotectant selection, dilution ratio, as well as freezing and thawing rates are parameters that interact with one another, and have all been found to vary greatly between species (Suquet et al., 2000; Rideout, Trippel & Litvak, 2004).

Recent research has shown that lumpfish (Cyclopterus lumpus), an epibenthic-pelagic fish species naturally found in the North Atlantic Ocean, is an effective cleaner fish in combating infestation with sea lice (Lepeophtheirus salmonis and Caligus elongats) among farmed salmon (Imsland et al., 2014), which is a growing problem in that industry (Costello, 2009). Commercial production of lumpfish juveniles is ongoing in Norway (Willumsen, 2001; Schaer & Vestvik, 2012), and on the Faroe Islands our aim is to establish a year-round production of lumpfish juveniles.

The lumpfish breeding season is considered to range from March to August (Kennedy et al., 2014), but in Faroese waters, mature lumpfish have been caught as late as in November (own observation). During the breeding season, brood fish are obtained along the coast lines, and are primarily captured by local fishing vessels using gillnets as the fish travel from deep waters towards the shore for spawning (Holst, 1993; Muus, 1999). Males specifically, are found around the shores during the breeding season as they prepare their territory and wait for females to arrive, so spawning can take place (Goulet, Green & Shears, 1986; Kennedy et al., 2014). Females lay their eggs in a nest site and then leave the spawning grounds for the males to stay and protect their offspring from predators (Muus, 1999; Kennedy et al., 2014). It has been reported that lumpfish males have a breeding period that lasts for 6–10 weeks (Davenport, 1985).

Reddish colouration is an indication that males have reached sexual maturity (Kells & Carpenter, 2011). Our observation is however, that this is not always a guarantee that stripping is successful. Additionally, stripping most often results in small volumes of milt: in our trials volume ranged from 0.08 to 3.2 ml. This is a limiting factor when a large number of eggs from females are to be fertilized at the same time. Delaying stripping of females is unadvisable as eggs may become overripe or the female may release roe before artificial fertilisation can be carried out (e.g., at night), resulting in unfertilised roe (own observation). Cryopreservation of lumpfish milt could solve the issues mentioned above and ensure a year-round supply of male gametes for a sustainable juvenile production. Additionally, storing of milt from individuals with desirable genetic qualities may be important if breeding programmes are to be initiated for this species.

In a pilot study (Raw data), we tested three cryosolutions (see Table S1) previously found successful in the cryopreservation of milt from other marine fish species, including Atlantic haddock (Melanogrammus aeglefinus) and Atlantic cod (Gadus morhua) (Rideout, Trippel & Litvak, 2004), Olive flounder (Paralichthys olivaceus) (Zhang et al., 2003) and Atlantic halibut (Hippoglossus hippoglossus) (Ding et al., 2011). We found that out of the three solutions, only spermatozoa cryopreserved in potassium bicarbonate and sucrose; the modified Mounib’s medium with 10% DMSO (Dimethyl sulfoxide), as used by Ding et al. (2011), resulted in motile spermatozoa post cryopreservation (see Table S2). The original Mounib’s medium also includes 6.5 mM reduced L-Glutathione (GSH) (Mounib, 1978), which is known to prevent free radicals from oxidising the lipid bilayer in the cell membrane, and thus prevents cell degradation. The role of sucrose in the medium is to stabilise the liposomal membrane of spermatozoa during cryopreservation (Quinn, 1985; Gwo, 2000).

Freezing rate has an effect on motility post-cryopreservation (Suquet et al., 2000). The height of the tray used in the cryopreservation process has a great effect on the rate. Rideout, Trippel & Litvak (2004) found that a 3 cm high tray resulted in samples reaching −90 °C in 90 s whereas it took 12 min on a tray 5.5 cm high. They also found better survival in the spermatozoa when the freezing rate was slow. In accordance with this, our pilot study indicated better motility post cryopreservation when samples were cryopreserved on a 4.8 cm tray compared to a 2.5 cm tray (Table S2).

Thawing temperature may also affect post-cryopreservation motility. For marine fish species, applied thawing temperatures typically vary between 10 and 40 °C (Suquet et al., 2000), these are lower than temperatures used for milt samples from freshwater fish species, which vary between 30 and 80 °C (Suquet et al., 2000). These findings correspond well to our pilot study where two thawing rates were tested; 37 °C and 50 °C and results showed clearly that lumpfish milt thawed in 37 °C resulted in higher motility scores (Table S2).

Our pilot study was unfortunately only based on milt samples from one lumpfish (stripped in 2012), so there were no replicates to base any firm conclusions on. Additionally, we did not have optimal equipment (cryo vials instead of straws) and were unable to carry out computer assisted sperm analysis (CASA).

Based on pilot study findings detailed above, we tested the efficacy of cryopreserving lumpfish milt (stripped in 2014) in three different diluents, all being variations based on Mounib’s medium (100 millimolar (mM) potassium bicarbonate (KHCO3), and 125 mm sucrose) as earlier findings indicated its success. We also tested the effect of two different freezing tray heights (4.8 and 6.4 cm). Motility performances of fresh and cryopreserved spermatozoa were analysed using a modern CASA system. Fertility post-cryopreservation was compared to the fertility of fresh milt.

Materials & Methods

Experimental design

Stripped milt from seven lumpfish males (weight: mean ± SD; 459.53 ± 348.10 g, and length: 22.44 ± 5.21 cm) was used in this experiment (some males were stripped more than once, so n = 16 samples). Stripping volume ranged from 0.3 to 2.7 ml per sample (0.97 ± 0.73) (Table S3). Motility percentage, pH (6.55 ± 0.21), osmolality (0.463 ± 0.06 mOsmol kg−1), and milt concentration (31.44 × 109 ± 8.35 × 109 cells ml−1) were measured after stripping each male. In the cryopreservation experiment, milt samples (n = 16) from seven fish were cryopreserved in triplicates in the following diluents: (1) Mounib’s medium without GSH “Mounib no GSH,” as in Ding et al. (2011), (2) Mounib’s original medium (Mounib, 1978) “Mounib,” and (3) Mounib’s medium as in “Mounib no GSH” but with added hen’s egg yolk (EY), which is a non-penetrating cryoprotectant that is commonly used in cryopreservation (Jamieson, 1991) “Mounib plus EY.” The cryoprotectant DMSO (10%) was used with all three diluents (Table 1). Additionally, we test two tray heights: (1) the previously successful 4.8 cm tray and, (2) a higher tray of 6.4 cm to ascertain whether an even slower freezing rate may further increase motility. All milt samples were tested at two freezing rates except “Mounib,” which was only tested at one of these (tray height: 4.8 cm) due to insufficient volumes of milt. For every milt sample used in the cryopreservation experiment, a matching sample of fresh milt (obtained from the same stripping) served as a control.

Table 1 Components included in cryopreserved samples.

The chemical composition of the three cryosolutions tested in this experiment. All are based on Mounib’s basic medium with and without reduced l-glutathione and hen’s egg yolk. Mounib no GSH and Mounib plus EY were tested on two freezing trays (height 4.8 and 6.4 cm), and Mounib was only tested on one tray (height 4.8 cm). The concentration in every sample was: milt to diluent (1:2) and 10% of the diluent volume was DMSO.

Cryosolution	Diluent	Dilution	Cryoprotectant	
Mounib	100 mM KHCO3, 125 mM Sucrose and 6.5 mM L-Gluthatione (reduced)	1:2	10% DMSO	
Mounib no GSH	100 mM KHCO3 and 125 mM Sucrose	1:2	10% DMSO	
Mounib plus EY	100 mM KHCO3, 125 mM Sucrose and 10% hen egg yolk	1:2	10% DMSO	
Notes.

All reagents in this table are purchased from VWR, Bie & Berntsen, Denmark.

Gamete collection

The majority of milt samples were obtained from fish produced from wild broodstock at Nesvík Marine Centre, Faroe Islands in 2013 (n = 12 samples from five fish). Other samples were obtained from wild fish captured near the shore by divers 8–10 days prior to this experiment (n = 4 samples from two fish) (Table S3). Fish were held in 3 m diameter cylindrical tanks, water depth 1 m with flow through filtered and UV treated sea water (35‰). Fish were fed ad lib with 3 mm standard commercial fish feed, Atlantic salmon type: “Margæti” from Havsbrún, Faroe Islands. The feed contained 48% and 27% crude protein and fat respectively.

Males with a reddish appearance were placed in a tank along with females (one male/three females) at ambient temperature (9–10 °C) for a few days prior to stripping in order to enhance gamete production (Klokseth & Øiestad, 1999). A pre-stripping check was done by light pressure on the sides and abdomen of the males. If milt was released, fish were placed in a 20 L container with sea water along with 20 ml of Benzocaine (anaesthetic). Once fish were unconscious, the milt was collected into 5 ml syringes, and placed on ice immediately. The fish were placed back into the tank with continually flowing sea water to recover.

Eggs were obtained from wild females (n = 10) captured 8–10 days prior to our trials near Faroese coastlines by local fishing vessels. Females with large bulging abdomens were easily identified as carrying roe and swelling around the urogenital opening signalled imminent spawning. Before stripping began, females were anaesthetized in same procedures as with the males mentioned earlier. Eggs were stripped by placing 1–2 fingers into the urogenital opening to release the roe and applying mild pressure to the abdomen, allowing the roe to drain into cleaned measuring containers. The amount of stripped eggs ranged from 100 ml to 400 ml per fish. Stripped eggs were immediately stored at 4 °C until the fertilisation procedure was initiated, which was usually within an hour.

Spermatozoa were counted under a microscope (Leica DM1000 led) using a hemocytometer (thoma 0.1 mm) using standard counting protocols: milt was diluted 1–1,000 in a non activating medium (NAM) previously used in Fauvel et al. (1998) prior to counting. 10 µl of the dilution was loaded into both counting chambers of the hemocytometer and allowed to settle for 10 min before counting the spermatozoa. Cells were counted in five squares each in two chambers on the thoma cell counting chamber (not counting cells on the bottom and right edges) and multiplied up to a cell count per ml. The osmolality of milt was measured using a Gonotec Osmomat 030-D cryoscopic osmometer (Gonotec, Berlin, Germany). The pH value of stripped milt was measured using a PHM 62 standard pH meter.

Motility measurements

Triplicates of fresh milt samples were examined within 30 min after stripping. The milt was diluted 1–200 in an activating medium (AM) made of 50% filtered sea water (SW), and 50% bovine serum albumin (BSA) (VWR, Bie & Berntsen, Denmark), prepared in distilled water (10 mg/ml) beforehand, to avoid the cells from sticking to the microscope slide. Immediately after milt was added to the AM, the dilution was cautiously mixed with the pipette tip to distribute the cells evenly, and then 6 µl of the dilution was quickly transferred into one chamber of a Leja 2 chamber CASA microscope slide (SC-20-01-02-B; Leja Products B.V., Nieuw-Vennep, Netherlands).

A two-step dilution procedure is often performed for measuring motility in milt (Dreanno et al., 1997; Fauvel et al., 1999; Groison et al., 2010). This procedure involves first diluting milt in NAM, an isotonic medium similar to the chemical composition of milt (Fauvel et al., 1998), in order to keep spermatozoa quiescent. Subsequently, the milt dilution is transferred to a microscope slide, where an activating agent (usually sea water) is added before monitoring the spermatozoa. In our trials, this procedure resulted in an uneven distribution of spermatozoa when the AM was added to the milt dilution within the chamber of the microscope slide. As lumpfish spermatozoa may be motile up to several minutes (G Norðberg, 2014, unpublished data), we chose to do a one-step procedure and diluted all samples in the AM directly, which allowed us to get an even distribution of cells for observation and recording video for the CASA system. G

All samples were observed with phase contrast (PH2) under a Leica DM1000 led microscope (object lens: 20×). A digital camera (Leica DFC 295) was attached to the microscope and coupled to a computer, and with the included Leica application suit (LAS) software, a clear live video feed of spermatozoa was obtained. Settings on the LAS software were adjusted to: 44.5 ms exposure, 1.4× gain and gamma 1.34, and image set to greyscale. Recording was achieved using a Blueberry software (BB flashback Pro 4) player. Two minutes of each sample was recorded, and the first 20 s of each movie were always excluded to avoid measurements of spermatozoa moving due to flow and avoiding the lag period caused by mixing milt and AM, and loading into the chamber of the slide. A total of 46 video frames were extracted from each video and saved as AVI files. Image J (http://rsb.info.nih.gov/ij) (Rasband) open source software, including a CASA plug-in that allows measuring the motility percentage of fish milt (Wilson-Leedy & Ingermann, 2007), was used for video analysis. To get accurate measurements of motility percentage of lumpfish spermatozoa using the CASA plug-in in ImageJ, the image of the imported AVI file threshold was adjusted to 57. In the CASA plug-in sperm tracker fields we only adjusted a few settings to get accurate measurements, these included, the maximum sperm size to 99 pixels, the minimum track length to 10 frames, the maximum sperm velocity between frames to 50 pixels, and the frame rate to 10 frames per seconds (the video was recorded at 10 fps).

Cryopreservation and thawing

“Mounib” and “Mounib no GSH” diluents were prepared within a week before the cryopreservation experiments, and stored at 4 °C along with hen’s egg yolk. DMSO was stored at room temperature. Milt samples in syringes were stored on ice not more than an hour before the cryopreservation set-up was prepared. The cryopreservation set-up was prepared on ice; first by adding diluent stock into cooled eppendorf tubes, thereafter DMSO. In cryosolution “Mounib plus EY,” egg yolk was added after the diluent, then the cryoprotectant. Finally, milt was added to all tubes, and ingredients cautiously mixed by aspiration with a pipette, samples were allowed to stand for a 10 min equilibration time. All samples were cryopreserved in 250 µl cryo-straws (Cryo Bio System, L’Aigle, France). Samples were drawn into the straws manually and sealed. They were then attached on to the top of a floating tray that was either 4.8 cm in height, or 6.4 cm in height, representing two freezing rates. Floating trays were placed within a Styrofoam box (inside dimensions H × L × W = 21 cm × 35.5 cm × 23 cm) filled with liquid nitrogen (−196 °C) with an approximate depth of 10 cm to allow straws to cool in the nitrogen gas layer for 10 min. Thereafter, the trays were turned over and samples plunged directly into the liquid nitrogen and left for at least 15 min before the thawing process was initiated. Straws were taken directly from liquid nitrogen into a water bath at 37 °C for a duration of 7 s to be thawed. The ends of straws were cut off with scissors to allow the samples to drain into fresh cooled eppendorf tubes. The examination of the cryopreserved milt was done in the same way as with fresh milt samples, only difference being that cryopreserved samples, having already been diluted in cryosolution, were diluted only 1–30 in the AM.

Fertilization

Lumpfish eggs become sticky and attach to each other on contact with sea water and form dense lumps within a few minutes (Davenport, 1985). This property can make it difficult to obtain good estimates of fertilisation success as visually inspecting roe on the inside of such lumps is impractical. Ballan Wrasse (Labrus bergylta) roe have a similar biochemical reaction when exposed to sea water, but mixing their roe with subtilisin, an enzyme also referred to as ‘alcalase’, along with Ringer’s solution removes the egg stickiness and allows eggs to remain unattached (Lein, Yoav & Tveiten, 2013). This method was tested in our own laboratory using lumpfish eggs with successful results (unpublished data) and was therefore incorporated into the fertilization procedure of this study. Before fertilization was initiated, the Ringer’s solution found in Young (1933) was combined with 10% Alcalase® 2.4 L FG (Novozymes A/S, Bagsværd, Denmark). Each sample (20 ml; about 2.300 eggs) was fertilised with milt at an approximately 1:30.000 egg to motile spermatozoa ratio. After adding the milt sample to the eggs, approximately 10 ml of sea water was stirred in to activate the spermatozoa. The compounds were mixed by gently stirring for at least 30 s, and then allowed to stand for 2 min before 10 ml of the Ringer’s + alcalase solution was added, and again gently stirred into to the sample for a few seconds to obtain an even distribution. The mixture was allowed to stand for 5 min before being poured into a fresh petri-dish containing SW (35‰), and gently stirred for a few seconds to dissolve the “glue” away from the eggs. Fertilized eggs were then placed in round incubators (15 L) with eight compartments separated by netting (1 mm mesh size) Water flow was set to 9 L ⋅ min−1 to allow enough water change in all compartments. Water temperature was measured daily and ranged between 11.1 and 11.3 °C. After 78.4–89.7 day degrees (dd), eggs were inspected under a dissection microscope to ascertain fertilisation. Approximately 50–200 eggs were inspected in each sample.

Statistical analysis

Analysis was carried out using R version 3.0.3 (R Core Team , 2014) and Excel (MS Office). While motility and fertilisation data are measured in percentages, the residuals in the chosen linear mixed effects models were sufficiently normal (Shapiro–Wilks tests for normality: Effect of cryopreservation on motility; W = 0.98, P = 0.41, Effect of tray height and diluent type on motility; W = 0.97, P = 0.36, Effect of cryopreservation on fertility; W = 0.96, P = 0.34), so using generalized linear models or transforming the data was not justified in this case.

Effect of cryopreservation and the effects of diluent type and tray height on motility were analysed using similar models. In both cases, a linear mixed effects model, “lmer” in the R package lme4 (Bates et al., 2014) was used with the aforementioned dependent and independent variables as well as “batch” as a random effect. Batch refers to an individual stripping of milt and any one male may produce several batches of milt. We chose “batch” as the random effect because males did not differ significantly in motility scores (Anova; F6,9 = 2.06, P = 0.16), which can be explained by high within-male variation in motility (Fig. S1). P values were extracted using a type II Wald Chi squared test included in the R package “car” (Fox & Weisberg, 2011). Effect of cryopreservation of milt on fertilisation success was analysed using a simple linear model.

Differences between treatments are expressed in percentage points (pp) and means are provided with Standard Deviations.

Ethical statement

As there is no animal experimentation legislation on the Faroe Islands, the local “animal protection act” was adhered to (Vinnumálaráðið, 1990) throughout this study. A fish veterinarian advised on best practice in relation to stripping to ensure no undue suffering caused by the procedure. Our impact on wild populations was limited, as we used predominately captive bred fish, which were bred for the purpose of producing a domestic lumpfish stock. There were no fish mortalities caused by our study and effort was put into providing optimal care and welfare for all fish involved.

Results

Cryopreserved milt had a 24.1 pp lower motility than fresh milt (type II Wald test; Chi squared = 41.88, df = 1, P < 0.001) with an average motility of fresh milt being 72.82 ± 13.60%, and that of cryopreserved milt 48.73 ± 19.01%. On average, milt samples treated with Mounib no GSH (tray height: 4.8 cm) had the best motility recovery of all treatments, which was 27.4 pp higher than milt samples cryopreserved with Mounib plus EY (tray height 6.4 cm), which had the lowest motility recovery.

Tray height had a significant effect on post-thawing motility with the lower tray (4.8 cm) producing a 13.0 pp higher motility than the higher one (Chi squared = 14.05, df = 1, P < 0.001; Fig. 1). Using different cryosolutions also affected the motility recovery; adding ‘EY’ to the modified Mounib’s medium resulted in a 19.3 pp lower motility while the modified Mounib’s medium had 6.4 pp higher motility than the original Mounib’s medium (Chi squared = 44.05, df = 2, P < 0.001) (Fig. 1).

Figure 1 Motility of cryopreserved sperm.

Mean motility of milt preserved in three different mediums on two different tray heights. Error bars represent 1 standard deviation.

The fertilisation success of eggs fertilised with fresh milt was not significantly higher than those fertilised with cryopreserved milt (30.81 ± 16.87% and 27.30 ± 17.02% respectively; F1,24 = 0.91, P = 0.35) (Table S4).

Discussion

This paper details a successful method for cryopreserving lumpfish milt. To our knowledge, no such methodology has previously been published on this species, and this information may be beneficial in future development of lumpfish as an aquaculture species. Our results indicate a motility loss of less than 14.2 pp when using our most optimal cryopreservation protocol. This is comparable to cryopreservation of e.g., cod, halibut, ocean pout, sea bream, striped trumpeter and turbot reviewed in Suquet et al. (2000), in which post-frozen motility recovery, expressed as a percentage of fresh sperm motility rate, ranged from 39 to 85% compared to our result, which was 80.92% at optimal tray height and cryosolution (67.8% on average).

Based on previous findings in our pilot study, we tested three variants of Mounib’s medium. Our results indicate that the best diluent is Mounib’s no GSH. This suggests that the penetrating cryoprotectant DMSO is sufficient to protect the spermatozoa from cell damages that occur during the freezing and thawing process. One would expect that the addition of GSH would have a positive effect on motility recovery as it has been shown in other animal species that the freezing process leads to a significant reduction of GSH content in spermatozoa (Bilodeau et al., 2000; Gadea et al., 2004; Gadea et al., 2005). In 2011, a study was carried out to test the antioxidant effect in cryopreservation of fish spermatozoa by adding different antioxidants to cryosolutions including GSH. The outcome was that the addition of GSH and any other antioxidant compound had no significant effect on motility or fertility, and in some instances it had a slightly negative effect (Lahnsteiner, Mansour & Kunz, 2011). This is similar to our findings, but a more recent study also using milt from rainbow trout (Oncorhynchus mykiss) revealed that addition of GSH (using the same concentration as the study from 2011) to the cryosolutions actually did have an significant positive effect on motility but no effect on fertility (Kutluyer et al., 2014). It is possible that lumpfish milt benefits from the antioxidant property of GSH during a longer storage period comparable to that in Mounib’s study (1978).

Our pilot study indicated that a tray height of 2.5 cm was too low, and findings in this study point to that 6.4 is too high with 13.04 pp lower motility than the lower tray (4.8 cm). Interestingly, Rideout, Trippel & Litvak’s (2004) 5.5 cm tray was better than their 3 cm tray. This suggests that there may be an optimal tray height yet to be found or that optimal tray height is species dependent. Perhaps the optimal height for lumpfish milt lies somewhere between 4.8 and 5.5 cm.

In Suquet et al. (2000), the optimal thawing temperature for marine fish species is said to vary from 10 to 40 °C. We have only tested two thawing temperatures (37 °C and 50 °C) in our pilot study, which showed that 37 °C is useful and 50 °C is not (Table S2). The duration time was 1 min or 1.5 min in that case, as samples were cryopreserved in vials, which need a longer thawing period, compared with straws as they are thinner and have a larger surface area. Therefore 37 °C was used in this study and the thawing time (7 s) was estimated to end when bubbles appeared within the straw—indicating that samples were thawed. It could therefore be interesting to test if an even lower thawing temperature and using a longer thawing time would optimize the motility recovery of lumpfish milt. It is also possible that tray height and thawing temperature interact, in which case a tray height resulting in low motility recovery at one temperature may result in better recovery at a different temperature.

This study showed that cryopreserved lumpfish milt can be used to fertilise eggs. Using our cryopreservation method the fertilisation success was only 3.51 pp lower using cryopreserved milt compared to fresh milt. The two groups were not significantly different, but our estimates are confounded by a number of factors that are not easily accounted for in this study. Female egg quality may differ drastically and thus fertilisation success (own observations), so variation seems to be large. Additionally, there are concerns regarding the ability to detect fertilised eggs early on (78–89 dd) as our own observations indicate a variability in speed of development. Because of this, some eggs that were in fact fertilised may have been classified as not fertilised. That being said, this kind of variation applies to both eggs fertilised with cryopreserved milt and fresh milt, so there was no systematic difference between the groups other than the milt used. In order to get good estimates of fertilisation success, we recommend obtaining a much larger sample size from one single population of fish.

Establishing a captive lumpfish broodstock for a production of cleanerfishes with certain genetic qualities may become essential in the future. Being able to cryopreserve lumpfish milt is a necessary part of this process, and we believe we have developed a reliable method, which can be put into use in most laboratories. It is also important to note that cryopreservation of lumpfish milt according to this protocol can be achieved without use of expensive equipment. These findings and this protocol will contribute to a reliable year-round production of lumpfish juveniles, and improve the utilization of the limited amounts of lumpfish milt available. Ultimately, this could allow for the generation of more in-depth knowledge and use of this fish species as a biological solution to sea-lice problems, without overfishing of the wild lumpfish population.

In conclusion: cryopreservation of lumpfish milt prepared in modified Mounib’s medium (Mounib no GSH) in a 1:2 (milt:diluents) dilution along with 10% DMSO is currently in use at our own hatchery, using the lower freezing rate (trey height: 4, 8 cm) and thawing procedure: 37 °C for 7 s. This study finds this to be the best combination of parameters, but the authors acknowledge that the method may be even further improved upon in further studies.

Supplemental Information

Table S1 Pilot study cryosolutions

Three different cryosolutions were tested in the pilot study. The chemical composition of the diluents and cryoprotectants are shown in the table along with the dilution factor (milt:diluent) used. The volume of every cryoprotectant was determined based on the volume of the diluents loaded.

Click here for additional data file.

Table S2 Motility results from pilot study

Milt cryopreserved in three different cryosolutions were tested with the first freezing and thawing procedure (trey height: 2.5 cm; thawed at 50 °C for 1 min). Only spermatozoa cryopreserved in cryosolution 2 did have any motility recovery, and therefore only spermatozoa cryopreserved in this solution were tested with the second freezing and thawing procedure (trey height: 4.5 cm; thawed at 37 °C for 1.5 min). Motility of fresh milt was measured before the freezing and thawing procedures and was evaluated a motility score of 4 (moving cells: 76–100%).

Click here for additional data file.

Table S3 Motility of fresh milt

Motility percentage of fresh milt and volume in ml (Vol.) obtained after each stripping (Batch) from individual lumpfish (Male) used in this study either captured wild (W) or from own brood stocks (B).

Click here for additional data file.

Table S4 Fertilisation results

Male batch (milt), female batch (roe), motility (mot. %) and fertilization (fert. %) data of roe fertilized with either fresh (F) or cryopreserved (C) milt. Cryopreserved milt in this table is treated with Mounib no GSH using trey height 4.8 cm. The average motility recovery (mean ± SD) of fresh milt is 67.20 ± 16.16 and 53.45 ± 18.55 for cryopreserved milt. The average fertilization success using fresh milt is 30.81 ± 16.87 and using cryopreserved milt 27.30 ± 17.02. Fertilization success using fresh milt is not significantly higher than using cryopreserved milt (F1,24 = 0.91, P = 0.35).

Click here for additional data file.

Figure S1 Motility by male

Motility of fresh milt split by males given as mean motility % ± SD.

Click here for additional data file.

Supplemental Information 6 Pilot study methods

This is a short write-up of the methods used in our pilot study. Some references are given, that are perhaps not included in the main manuscript, so there is a reference list included in the pilot study methods.

Click here for additional data file.

Raw data These are the data used for the result secction of the main manuscript. There are two sheets; cryopreservation of milt and fertilisation success.

Click here for additional data file.

Authors would like to thank the staff at Nesvík Marine Centre, Faroe Islands for their contribution in this study. As well as the staff at Føroya Sjósavn (The National Aquarium in Tórshavn, Faroe Islands.) for helping with the capture of wild lumpfish males. Also we would like to thank Anne Laure Groison for helpful guidance, especially concerning some of the equipment used in this study.

Additional Information and Declarations

Competing Interests

Author Contributions

Animal Ethics

All authors are employees at Fiskaaling, Aquacultural Research Station of the Faroese.

Gunnvør Norðberg conceived and designed the experiments, performed the experiments, analyzed the data, contributed reagents/materials/analysis tools, wrote the paper, prepared figures and/or tables, reviewed drafts of the paper.

Asa Johannesen analyzed the data, contributed reagents/materials/analysis tools, prepared figures and/or tables, reviewed drafts of the paper.

Regin Arge contributed reagents/materials/analysis tools, reviewed drafts of the paper.

The following information was supplied relating to ethical approvals (i.e., approving body and any reference numbers):

As there is no animal experimentation legislation on the Faroe Islands, the local “animal protection act” was adhered to (Løgtingslóg nr. 9 frá 14. mars 1985 um vernd av dýrum / værn af dyr, sum seinast broytt við løgtingslóg nr. 60 frá 30. mai 1990. Available at: http://logir.fo/Logtingslog/9-fra-14-03-1985-um-vernd-av-dyrum–vaern-af-dyr-sum-seinast-broytt) throughout this study. A fish veterinarian advised on best practice in relation to stripping to ensure no undue suffering caused by the procedure. Our impact on wild populations was limited, as we used predominately captive bred fish, which were bred for the purpose of producing a domestic lumpfish stock. There were no fish mortalities caused by our study and effort was put into providing optimal care and welfare for all fish involved.

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
