# Peer review of "Cryopreservation of lumpfish Cyclopterus lumpus (Linnaeus, 1758) milt"

_PeerJ, doi:10.7717/peerj.1003_

## Round 0.1 · original submission · Major Revisions

· Academic Editor

Major Revisions

The ms lacks precision in reporting Materials and Methods and in presenting the Results, including expanlations of the statistical treatment. Please address the reviewers comments.

·

Basic reporting

To the best of my knowledge, the manuscript adheres to the PeerJ policies. The manuscript is written in clear English, and contains sufficient information in the introduction to demonstrate why the research is important and prepare the reader for the contained study. The manuscript has a standard structure style.

The figures are understandable, but lack information in the figure captions, such as the unit and variance for data that are presented (i.e. are they mean +- SD?)

The manuscript is self-contained, but ideally should have more results. For instance, the fertilisation success of stripped eggs with milt cryopreserved with different methods would provide the ultimate proof of a usable protocol. Comments are made in this regard in the discussion, so is it possible to provide some numbers in this regard?

Experimental design

All criteria are fullfilled

Validity of the findings

The findings presented are robust within context (see prior comment regarding fertilisation success). They appear statistically sound, and have appropriate controls.

Due to the non-conventional statistics employed to help normalise for batch to batch differences between milt, the results section does read as "statistically heavy" and could be simplified to make it more understandable to a wider audience. For instance stating the T values of the R summary model for different parameters (lines 205-207) requires more explanation as many people will not be familiar with the R program, i.e. what does the tray: -3.75 T value mean?

The discussion is short and has space to discuss the underlying reasons for why, for instance, GSH may have negatively affected sperm motility, did it for instance make the preservation solution too reductive. Is there a possibility that over long term storage, such as that which was originally performed by Mounib, lumpfish sperm may have benefited from the antioxidant capacity of GSH?

Additional comments

Further comments in regard to minor issues are included in the annotated pdf attachment

Reviewer 2 ·

Basic reporting

The introduction generally covers the topic investigated, and has included sufficient and updated references. However, the introduction could be slightly rewritten. From line 52 a pilot study is reported where the data obtained is used to focus this study. However, conclusions are drawn without actually presenting any data, and without showing the design of the pilot study (was it the same males that were tested for the different solutions, how many, which temperatures etc.). Some of these results are important, as “only spermatozoa cryopreserved in the modified Mounib’s medium resulted in motile sperm”. You should either consider adding more information (e.g. in supplementary data) or omit presenting the data in the introduction. Table 1 could easily be included in the text.
Line 43: should it be stripping of females? And, when you are writing “Delaying stripping or females is unadvisable as they may release their roe prematurely.”, this seems a bit contradictory, normally delaying stripping will lead to overripe eggs.
Line 49-51, wording is odd. If a successful protocol already has been established, why this experiment? If you refer to this manuscript, then rephrase.
The authors should use full common names (e.e. Atlantic haddock) and add latin names.
The figures is not up to expected standards. None of them explains the boxes in the figure, or what the numbers represent. The numbers in the y-aksis should has an angle?, and the axis title is different between the figures, and both uses Motility as % while in the text you use pp based on models? Figure 1 could easily be included in the text. Results as “suggest” should be in text and not figure.
Line 73- 79. You have actually managed to write ALL references in this section erroneous one way or the other! Ding et al; 2011 should be Ding et al. 2011, Mounib, 1987 should be Mounib 1987, Ding et al (2011) should be Ding et al. (2011), Jamieson, 1991 should be Jamieson 1991.

Experimental design

Generally, you add to little information of individual males. You could replace Fig. 1 with a table showing the results for the individual males and their portions, and whether these are farmed or wild. This would allow us to judge about the variation in the results, Fig. 1 (if this is max-min) indicates large variation, and without any notice about whether the ranking of results is the same for fresh and preserved.

I understand that the limited number of batches obtained from the males makes it difficult to compare males? But I do not understand what you mean by that “each batch differed sufficiently that “male” was not a useful random batches”. Does this mean thay You did compare the males? Also, why is not male origin included (wild, farmed) in the model.

Line 100: Please add actual numbers of fish and milt samples for farmed and wild fish
Line 105: Add rearing temperature
Line 104: Add feed type and supplier
Line 114: What is standard counting procedures? How many/much were counted osv.
Line 185: Whys have you not used actual values instead of a model, and then how have you calculated mean etc?
Line 196 and Fig. 1: You must write that these are not actual values but modeled. But why do you not use “real” values?
Figure 2. Legends states “Mounib – L-Glu”, figure text “Mounib no L-Glu”

Validity of the findings

As stated above, more data should be added for the males.
The language in the discussion is not good, e.g. “and may this information be beneficial in future” could be written “and this information may be beneficial in future
Line 215: Concordant?
Line 229 – 237 is introduction.
Generally, the discussion is short and to the point. However, you have actually not tested whether the cryopreserved milt is able to fertilize eggs (in this experiment at least), and the part regarding the benefits using this procedure should be written with this in mind.

Additional comments

This should be a very short article, could be much shorter and to the point than it is written. Many conclusions is drawn from the pilot experiment without presenting any data. The MM does not include all relevant data, data regarding males should be included. Results lacks comparisions of males, comparisions between farmed and wild fish. Results are presented as pp based on modelled values instead of actiually measured % values, this is not mentioned in the figures. The optimal test, that cryopreserved milt functions is not tested or emphasised.

Reviewer 3 ·

Basic reporting

The English language should be improved by the help of someone who knows the language well. Some of the sentences and statements could be simplified, shortened and more precise.
The authors need to consult the templates for the journal, for example should the abstract be placed between the author’s names and the introduction. Also, the references ought to be laid out in agreement with the guidelines.
Line 73-79 in the Introduction to my opinion belong under Material and methods.
Material/methods: The number stripped for milt once or twice should be given (line 100).
Two authors are missing in the Reference list: Jamieson, 1991 (Introduction line 77) and Rayling et al. (Discussion, line 226).

Experimental design

No coments.

Validity of the findings

The data are valid and of importance for the industry. However, to my opinion, the statements that the procedure for cryopreservation of lumpfish milt is successful and will be of main importance for the development of the lumpfish production (Discussion, line 212 and 241, Abstract line 1) should be modified as long as no fertilization tests have been done.
Statistics: The models used for analyzing the data must be better described. Fixed and random effects must be shown for both models.
Possible effect of even lower thawing temperatures and duration of thawing should be considered when concluding about the results.

---

## Round 0.2 · Minor Revisions

· Academic Editor

Minor Revisions

This manuscript only needs very minor changes, as indicated by reviewer 3, before it can be accepted. Congratulations.

Reviewer 2 ·

Basic reporting

No comments

Experimental design

No comments

Validity of the findings

No comments

Additional comments

The latin names needs not be repeated in lines 64-68.

Reviewer 3 ·

Basic reporting

In line 21-22 you state that “Cryopreservation efforts for marine fishs pecies were ongoing already in the 90’s….”. Bolla et al. published results for cryopreservation of A. halibut milt already in 1987. This should probably be referred to. (Bolla, S., Holmefjord, I. & Refstie, T., 1987. Cryogenetic preservation of Atlantic halibut sperm. Aquaculture, volume 65, pp. 371-374.)
Line 99: I suggest “In the present experiment, based on…, we tested….” In the rest of the manuscript you use past, but in this paragraph only you use the present tense. Change to past tense for the whole paragraph.
Line 328: I would either say “our most optimal method”, or skip the word optimal.

Lin 332: I suggest “ to our result which was 90.92 when using the most optimal tray….”
Line 358. I suggest to skip the last sentence.

Line 361: “In Suceuet et al. (2000) the optimal thawing temperature…”
Line 371: I suggest to use the word result instead of produce.

Experimental design

The authors have now included fertilization data have been included in the manuscript. However, to me it is not clear what fresh milt represents. Was half the volume of milt frozen and half used as fresh milt, or did you strip the same males for fresh milt at a later stage? This needs to be made clear in Material and Methods.

Validity of the findings

The discussion has been imporved, and the conclusions are mor appropriately stated than in the original manuscript.

Additional comments

I think the information in this article is useful for both for scientists and for commercial production of lumpsucker. The manuscript is still rich in words, but I reccomend that it sholud be published with minor revision.

---

## Round 0.3 · accepted · Accept

· Academic Editor

Accept

I think your work will help establish commercial farming og lumpfish, which is at its very beginning. It will be helpful both to farmers and in research.